# Launch and Iterate: Reducing Prediction Churn

**Q. Cormier**
ENS Lyon
15 parvis René Descartes
Lyon, France
quentin.cormier@ens-lyon.fr

**M. Milani Fard, K. Canini, M. R. Gupta**
Google Inc.
1600 Amphitheatre Parkway
Mountain View, CA 94043
{mmilanifard,canini,mayagupta}@google.com

## Abstract

Practical applications of machine learning often involve successive training itera-
tions with changes to features and training examples. Ideally, changes in the output
of any new model should only be improvements (wins) over the previous iteration,
but in practice the predictions may change neutrally for many examples, resulting
in extra net-zero wins and losses, referred to as unnecessary churn. These changes
in the predictions are problematic for usability for some applications, and make it
harder and more expensive to measure if a change is statistically significant positive.
In this paper, we formulate the problem and present a stabilization operator to regu-
larize a classifier towards a previous classifier. We use a Markov chain Monte Carlo
stabilization operator to produce a model with more consistent predictions without
adversely affecting accuracy. We investigate the properties of the proposal with
theoretical analysis. Experiments on benchmark datasets for different classification
algorithms demonstrate the method and the resulting reduction in churn.

## 1 The Curse of Version 2.0

In most practical settings, training and launching an initial machine-learned model is only the first
step: as new and improved features are created, additional training data is gathered, and the model
and learning algorithm are improved, it is natural to launch a series of ever-improving models. Each
new candidate may bring wins, but also unnecessary changes. In practice, it is desirable to minimize
any unnecessary changes for two key reasons. First, unnecessary changes can hinder usability
and debugability as they can be disorienting to users and follow-on system components. Second,
unnecessary changes make it more difficult to measure with statistical confidence whether the change
is truly an improvement. For both these reasons, there is great interest in making only those changes
that are wins, and minimizing any unnecessary changes, while making sure such process does not
hinder the overall accuracy objective.

There is already a large body of work in machine learning that treats the stability of learning
algorithms. These range from the early works of Devroye and Wagner [1] and Vapnik [2, 3] to more
recent studies of learning stability in more general hypothesis spaces [4, 5, 6]. Most of the literature
on this topic focus on stability of the learning algorithm in terms of the risk or loss function and how
such properties translate into uniform generalization with specific convergence rates. We build on
these notions, but the problem treated here is substantively different.

We address the problem of training consecutive classifiers to reduce unnecessary changes in the
presence of realistic evolution of the problem domain and the training sets over time. The main
contributions of this paper include: (I) discussion and formulation of the "churn" metric between
trained models, (II) design of stabilization operators for regularization towards a previous model, (III)
proposing a Markov chain Monte Carlo (MCMC) stabilization technique, (VI) theoretical analysis of
the proposed stabilization in terms of churn, and (V) empirical analysis of the proposed methods on
benchmark datasets with different classification algorithms.

Table 1: Win-loss ratio (WLR) needed to establish a change is statistically significant at the $p = 0.05$ level for $k$ wins out of $n$ diffs from a binomial distribution. The empirical WLR column shows the WLR one must actually see in the diffs. The true WLR column is the WLR the change must have so that any random draw of diffs has at least a 95% chance of producing the needed empirical WLR.

| # Diffs | Min # Wins Needed | Max # Losses Allowed | Empirical WLR Needed | True WLR Needed |
|---|---|---|---|---|
| 10 | 9 | 1 | 9.000 | 26.195 |
| 100 | 59 | 41 | 1.439 | 1.972 |
| 1,000 | 527 | 473 | 1.114 | 1.234 |
| 10,000 | 5,083 | 4,917 | 1.034 | 1.068 |

## 1.1 Testing for Improvements

In the machine learning literature, it is common to compare classifiers on a fixed pre-labeled test set. However, a fixed test set has a few practical downsides. First, if many potential changes to the model are evaluated on the same dataset, it becomes difficult to avoid observing spurious positive effects that are actually due to chance. Second, the true test distribution may be evolving over time, meaning that a fixed test set will eventually diverge from the true distribution of interest. Third, and most important to our discussion, any particular change may affect only a small subset of the test examples, leaving too small a sample of differences (*diffs*) to determine whether a change is statistically significant.

For example, suppose one has a fixed test set of 10,000 samples with which to evaluate a classifier. Consider a change to one of the features, say a Boolean string-similarity feature that causes the feature to match more synonyms, and suppose that re-training a classifier with this small change to this one feature impacts only 0.1% of random examples. Then only 10 of the 10,000 test examples would be affected. As shown in the first row of Table 1, given only 10 diffs, there must be 9 or more wins to declare the change statistically significantly positive for $p = 0.05$.

Note that cross-validation (CV), even in leave-one-out form, does not solve this issue. First, we are still bound by the size of the training set which might not include enough diffs between the two models. Second, and more importantly, the model in the previous iteration has likely seen the entire dataset, which breaks the independence assumption needed for the statistical test.

To address these problems and ensure a fresh, sufficiently large test set for each comparison, practitioners often instead measure changes on a set of diffs for the proposed change. For example, to compare classifier $A$ and $B$, each classifier is evaluated on a billion unlabeled examples, and then the set of diffs is defined as those examples for which classifiers $A$ and $B$ predict a different class.

## 1.2 Churn

We define the *churn* between two models as the expected percent of diffs sampled from the test distribution. For a fixed accuracy gain, less churn is better. For example, if classifier $A$ has accuracy 90% and classifier $B$ has accuracy 91%, then the best case is if classifier $B$ gets the same 90% of examples correct as classifier $A$, while correcting $A$'s errors on 1% of the data. Churn is thus only 1% in this case, and all diffs between $A$ and $B$ will be wins for $B$. Therefore the improvement of $B$ over $A$ will achieve statistical significance after labelling a mere 10 diffs. The worst case is if classifier $A$ is right on the 9% of examples that $B$ gets wrong, and $B$ is right on the 10% of examples that $A$ gets wrong. In this case, churn is 19%, and a given diff will only have probability of 10/19 of being a win for $B$, and almost 1,000 diffs will have to be labeled to be confident that $B$ is better.

**On Statistical Significance**: Throughout this paper, we assume that every diff is independent and identically distributed with some probability of being a win for the *test* model vs. the *base* model. Thus, the probability of $k$ wins in $n$ trials follows a binomial distribution. Confidence intervals can provide more information than a p-value, but p-values are a useful summary statistic to motivate the problem and proposed solution, and are relevant in practice; for a longer discussion see e.g. [7].

## 2 Reducing Churn for Classifiers

In this paper, we propose a new training strategy for reducing the churn between classifiers. One special case is how to train a classifier $B$ to be low-churn given a fixed classifier $A$. We treat that

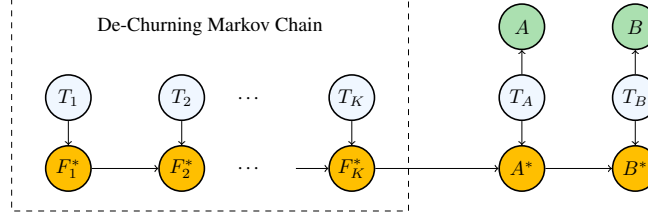

Figure 1: The orange nodes illustrate a Markov Chain, at each step the classifier $F_t^*$ is regularized towards the previous step's classifier $F_{t-1}^*$ using the stabilization operator $\mathcal{S}$, and each step trained on a different random training set $T_t$. We run $K$ steps of this Markov chain, for $K$ large enough so that the distribution of $F_k^*$ is close to a stationary distribution. The classifier $A^* = \mathcal{S}(F_K^*, T_A)$ is then deployed. Later, some changes are proposed, and a new classifier $B^*$ is trained on training set $T_B$ but regularized towards $A^*$ using $B^* = \mathcal{S}(A^*, T_B)$. We compare this proposal in terms of churn and accuracy to the green nodes, which do not use the proposed stabilization.

special case as well as a broader problem: a framework for training both classifiers $A$ and $B$ so that classifier $B$ is expected to have low-churn relative to classifier $A$, though when we train $A$ we do not yet know exactly the changes $B$ will incorporate. We place no constraints on the kind of classifiers or the kind of future changes allowed.

Our solution consists of two components: a *stabilization* operator that regularizes classifier $B$ to be closer in predictions to classifier $A$; and a *randomization* of the training set that attempts to mimic expected future changes.

We consider a training set $T = \{(x_i, y_i)\}_{i=1}^m$ of $m$ samples with each $D$-dimensional feature vector $x_i \in \mathcal{X} \subseteq \mathbb{R}^D$ and each label $y_i \in \mathcal{Y} = \{-1, 1\}$. Samples are drawn i.i.d. from distribution $\mathcal{D}$. Define a classifier $f : \mathbb{R}^D \to \{-1, 1\}$, and the churn between two classifiers $f_1$ and $f_2$ as:

$$C(f_1, f_2) = \mathop{\mathbb{E}}_{(X,Y) \sim \mathcal{D}} [\mathbb{1}_{f_1(X)f_2(X) < 0}], \tag{1}$$

where $\mathbb{1}$ is the indicator function. We are given training sets $T_A$ and $T_B$ to train the first and second version of the model respectively. $T_B$ might add or drop features or examples compared to $T_A$.

## 2.1 Perturbed Training to Imitate Future Changes

Consider a random training set drawn from a distribution $\mathcal{P}(T_A)$, such that different draws may have different training samples and different features. We show that one can train an initial classifier to be more consistent in predictions for different realizations of the perturbed training set by iteratively training on a series of i.i.d. random draws $T_1, T_2, \ldots$ from $\mathcal{P}(T_A)$. We choose $\mathcal{P}(T_A)$ to model a typical expected future change to the dataset. For example, if we think a likely future change will add 5% more training data and one new feature, then we would define a random training set to be a random 95% of the $m$ examples in $T_A$, while dropping a feature at random.

## 2.2 Stabilized Training Based On A Previous Model using a Markov Chain

We propose a Markov chain Monte Carlo (MCMC) approach to form a distribution over classifiers that are consistent in predictions w.r.t. the distribution $\mathcal{P}(T_A)$ on the training set. Let $\mathcal{S}$ denote a regularized training that outputs a new classifier $F_{t+1}^* = \mathcal{S}(F_t^*, T_{t+1})$ where $F_t^*$ is a previous classifier and $T_{t+1}$ is the current training set. Applying $\mathcal{S}$ repeatedly to random training sets $T_t$ forms a Markov chain as shown in Figure 1. We expect this chain to produce a stationary peaked distribution on classifiers robust to the perturbation $\mathcal{P}(T_A)$. We sample a model from this resulting distribution after $K$ steps.

We end the proposed Markov chain with a classifier $A^*$ trained on the full training set $T_A$, that is, $A^* = \mathcal{S}(F_K^*, T_A)$. Classifier $A^*$ is the initial launched model, and has been pre-trained to be robust to the kind of changes we expect to see in some future training set $T_B$. Later, classifier $B^*$ should be trained as $B^* = \mathcal{S}(A^*, T_B)$. We expect the chain to have reduced the churn $C(A^*, B^*)$ compared to the churn $C(A, B)$ that would have resulted from training classifiers $A$ and $B$ without the proposed stabilization. See Figure 1 for an illustration. Note that this chain only needs to be run for the first version of the model.

**On Regularization Effect of Perturbed Training:** One can view the perturbation of the dataset and random feature drops during the MCMC run as a form of regularization, resembling the dropout technique [8] now popular in deep, convolutional and recurrent neural networks (see e.g. [9] for a recent survey). Such regularization can result in better generalization error, and our empirical results show some evidence of such an effect. See further discussion in the experiments section.

**Perturbation Chain as Longitudinal Study:** The chain in Figure 1 can also be viewed as a study of the stabilization operator upon several iterations of the model, with each trained and anchored on the previous version. It can help us assess if the successive application of the operator has any adverse effect on the accuracy or if the resulting churn reduction diminishes over time.

## 3 Stabilization Operators

We propose two stabilization operators: (I) Regress to Corrected Prediction (RCP) which turns the classification problem into a regression towards corrected predictions of an older model, and (II) the Diplopia operator which regularizes the new model towards the older model using example weights.

### 3.1 RCP Stabilization Operator

We propose a stabilization operator $\mathcal{S}(f,T)$ that can be used with almost any regression algorithm and any type of change. The RCP operator re-labels each classification training label $y_j \in \{-1,1\}$ in $T$ with a regularized label $\tilde{y}_j \in \mathbb{R}$, using an *anchor* model $f$:

$$\tilde{y}_j = \begin{cases} \alpha f(x_j) + (1-\alpha)y_j & \text{if } y_j f(x_j) \geq 0 \\ \epsilon y_j & \text{otherwise,} \end{cases} \tag{2}$$

where $\alpha, \epsilon \in [0,1]$ are hyperparameters of $\mathcal{S}$ that control the churn-accuracy trade-off, with larger $\alpha$ corresponding to lower churn but less sensitive to good changes. Denote the set of all re-labeled examples $\tilde{T}$. The RCP stabilization operator $\mathcal{S}$ trains a regression model on $\tilde{T}$, using the user's choice of regression algorithm.

### 3.2 Diplopia Stabilization Operator

The second stabilization operator, which we term Diplopia (double-vision), can be used with any classification strategy that can output a probability estimate for each class, including algorithms like SVMs and random forests (calibrated with a method like Platt scaling [10] or isotonic regression [11]). This operator can be easily extended to multi-class problems.

For binary classification, the Diplopia operator copies each training example into two examples with labels $\pm 1$, and assigns different weights to the two contradictorily labeled copies. If $f(.)$ is the probability estimate of class $+1$:

$$(x_i, y_i) \rightarrow \begin{cases} (x_i, +1) \text{ with weight } \Lambda_i \\ (x_i, -1) \text{ with weight } 1 - \Lambda_i \end{cases} \quad \Lambda_i = \begin{cases} \alpha f(x_i) + (1-\alpha)\mathbb{1}_{y_i \geq 0} & \text{if } y_i(f(x_i) - \frac{1}{2}) \geq 0 \\ 1/2 + \epsilon y_i & \text{otherwise.} \end{cases}$$

The formula always assigns the higher weight to the copy with the correct label. Notice that the roles of $\alpha$ and $\epsilon$ are very similar than to those in (2). To see the intuition behind this operator, note that with $\alpha = 1$ and without the $\epsilon$-correction, stochastic $f(.)$ maximizes the likelihood of the new dataset.

The RCP operator requires using a regressor, but our preliminary experiments showed that it often trains faster (without the need to double the dataset size) and reduces churn better than the Diplopia operator. We therefore focus on the RCP operator for theoretical and empirical analysis.

## 4 Theoretical Results

In this section we present some general bounds on smoothed churn, assuming that the perturbation does not remove any features, and that the training algorithm is symmetric in training examples (i.e. independent of the order of the dataset). The analysis here assumes datasets for different models are sampled i.i.d., ignoring the dependency between consecutive re-labeled datasets (through the intermediate model). Proofs and further technical details are given in the supplemental material.

First, note that we can rewrite the definition of the churn in terms of zero-one loss:

$$C(f_1, f_2) = \mathop{\mathbb{E}}_{(X,Y)\sim\mathcal{D}} [\ell_{0,1}(f_1(X), f_2(X))] = \mathop{\mathbb{E}}_{(X,Y)\sim\mathcal{D}} [|\ell_{0,1}(f_1(X), Y) - \ell_{0,1}(f_2(X), Y)|]. \quad (3)$$

We define a relaxation of $C$ that is similar to the loss used by [5] to study the stability of classification algorithms, we call it *smooth churn* and it is parameterized by the choice of $\gamma$:

$$C_\gamma(f_1, f_2) = \mathop{\mathbb{E}}_{(X,Y)\sim\mathcal{D}} [|\ell_\gamma(f_1(X), Y) - \ell_\gamma(f_2(X), Y)|], \quad (4)$$

where $\ell_\gamma(y, y') = 1$ if $yy' \leq 0$, $\ell_\gamma(y, y') = 1 - yy'/\gamma$ for $0 \leq yy' \leq \gamma$, and $\ell_\gamma(y, y') = 0$ otherwise. Smooth churn can be interpreted as $\gamma$ playing the role of a "confidence threshold" of the classifier $f$ such that $|f(x)| \ll \gamma$ means the classifier is not confident in its prediction. It is easy to verify that $\ell_\gamma$ is $(1/\gamma)$-Lipschitz continuous with respect to $y$, when $y' \in \{-1, 1\}$.

Let $f_T(x) \to \mathbb{R}$ be a classifier discriminant function (which can be thresholded to form a classifier) trained on set $T$. Let $T^i$ be the same as $T$ except with the $i$th training sample $(x_i, y_i)$ replaced by another sample. Then, as in [4], define training algorithm $f_.(.)$ to be $\beta$-stable if:

$$\forall x, T, T^i : |f_T(x) - f_{T^i}(x)| \leq \beta. \quad (5)$$

Many algorithms such as SVM and classical regularization networks have been shown to be $\beta$-stable with $\beta = O(1/m)$ [4, 5]. We can use $\beta$-stability of learning algorithms to get a bound on the expected churn between independent runs of the algorithms on i.i.d. datasets:

**Theorem 1** (Expected Churn). *Suppose $f$ is $\beta$-stable, and is used to train classifiers on i.i.d. training sets $T$ and $T'$ sampled from $\mathcal{D}^m$. We have:*

$$\mathop{\mathbb{E}}_{T,T'\sim\mathcal{D}^m}[C_\gamma(f_T, f_{T'})] \leq \frac{\beta\sqrt{\pi m}}{\gamma}. \quad (6)$$

Assuming $\beta = O(1/m)$ this bound is of order $O(1/\sqrt{m})$, in line with most concentration bounds on the generalization error. We can further show that churn is concentrated around its expectation:

**Theorem 2** (Concentration Bound on Churn). *Suppose $f$ is $\beta$-stable, and is used to train classifiers on i.i.d. training sets $T$ and $T'$ sampled from $\mathcal{D}^m$. We have:*

$$\mathop{\Pr}_{T,T'\sim\mathcal{D}^m} \left\{ C_\gamma(f_T, f_{T'}) > \epsilon + \frac{\sqrt{\pi m}\beta}{\gamma} \right\} \leq e^{-\frac{\epsilon^2\gamma^2}{m\beta^2}}. \quad (7)$$

$\beta$-stability for learning algorithms often includes worst case bound on loss or Lipschitz-constant of the loss function. Assuming we use the RCP operator with squared loss in a reproducing kernel Hilbert space (RKHS), we can derive a distribution-dependent bound on the expected squared churn:

**Theorem 3** (Expected Squared Churn). *Let $\mathcal{F}$ be a reproducing kernel Hilbert space with kernel $k$ such that $\forall x \in \mathcal{X} : k(x, x) \leq \kappa^2 < \infty$. Let $f_T$ be a model trained on $T = \{(x_i, y_i)\}_{i=1}^m$ defined by:*

$$f_T = \arg\min_{g\in\mathcal{F}} \frac{1}{m} \sum_1^m (g(x_i) - y_i)^2 + \lambda\|g\|_k^2. \quad (8)$$

*For models trained on i.i.d. training sets $T$ and $T'$:*

$$\mathop{\mathbb{E}}_{\substack{T,T'\sim\mathcal{D}^m \\ (X,Y)\sim\mathcal{D}}} \left[ (\ell_\gamma(f_T(X), Y) - \ell_\gamma(f_{T'}(X), Y))^2 \right] \leq \frac{2\kappa^4}{m\lambda^2\gamma^2} \mathop{\mathbb{E}}_{T\sim\mathcal{D}^m} \left[ \frac{1}{m} \sum_{i=1}^m (f_T(x_i) - y_i)^2 \right]. \quad (9)$$

We can further use Chebyshev's inequality to get a concentration bound on the smooth churn $C_\gamma$. Unlike the bounds in [4] and [5], the bound of Theorem 3 scales with the expected training error (note that we must use $\tilde{y}_i$ in place of of $y_i$ when applying the theorem, since training data is re-labeled by the stabilization operator). We can thus use the above bound to analyse the effect of $\alpha$ and $\epsilon$ on the churn, through their influence on the training error.

Suppose the Markov chain described in Section 2.2 has reached a stationary distribution. Let $F_k^*$ be a model sampled from the resulting stationary distribution, used with the RCP operator defined in (2)

Table 2: Description of the datasets used in the experimental analysis.

|  | Nomao [13] | News Popularity [14] | Twitter Buzz [15] |
|---|---|---|---|
| # Features | 89 | 61 | 77 |
| $T_A$ | 4000 samples, 84 features | 8000 samples, 58 features | 4000 samples, 70 features |
| $T_B$ | 5000 samples, 89 features | 10000 samples, 61 features | 5000 samples, 77 features |
| Validation set | 1000 samples | 1000 samples | 1000 samples |
| Testing set | 28465 samples | 28797 samples | 45402 samples |

to re-label the dataset $T_{k+1}$. Since $F_{k+1}^*$ is the minimizer of objective in (8) on the re-labeled dataset we have:

$$
\begin{aligned}
\mathop{\mathbb{E}}_{T_{k+1}} \left[ \frac{1}{m} \sum_{i=1}^{m} (F_{k+1}^*(x_i) - \tilde{y}_i)^2 \right] &\leq \mathop{\mathbb{E}}_{T_{k+1}} \left[ \frac{1}{m} \sum_{i=1}^{m} (F_k^*(x_i) - \tilde{y}_i)^2 + \lambda (\|F_k^*\|_k^2 - \|F_{k+1}^*\|_k^2) \right] \\
&= \mathop{\mathbb{E}}_{T_{k+1}} \left[ \frac{1}{m} \sum_{i=1}^{m} (F_k^*(x_i) - \tilde{y}_i)^2 \right],
\end{aligned}
\tag{10}
$$

where line (10) is by the assumptions of stationary regime on $F_k^*$ and $F_{k+1}^*$ with similar dataset sampling distributions for $T_k$ and $T_{k+1}$. If $\mathcal{E}$ is the set of examples that $F_k^*$ got wrong, using the definition of the RCP operator we can replace $\tilde{y}_i$ to get this bound on the squared churn:

$$
\frac{\kappa^4}{m\lambda^2\gamma^2} \mathop{\mathbb{E}}_{T_{k+1}} \left[ \frac{1-\alpha}{m} \sum_{i \notin \mathcal{E}} (F_k^*(x_i) - y_i)^2 + \frac{1}{m} \sum_{i \in \mathcal{E}} (F_k^*(x_i) + \epsilon)^2 \right].
\tag{11}
$$

We can see in Eqn. (11) that using an $\alpha$ close to 1 can decrease the first part of the bound, but at the same time it can negatively affect the error rate of the classifier, resulting in more samples in $\mathcal{E}$ and consequently a larger second term. Decreasing $\epsilon$ can reduce the $(F_k^*(x_i) + \epsilon)^2$ term of the bound, but can again cause an increase in the error rate. As shown in the experimental results, there is often a trade-off between the amount of churn reduction and the accuracy of the resulting model. We can measure the accuracy on the training set or a validation set to make sure the choice of $\alpha$ and $\epsilon$ does not degrade the accuracy. To estimate churn reduction, we can use an un-labeled dataset.

## 5 Experiments

This section demonstrates the churn reduction effect of the RCP operator for three UCI benchmark datasets (see Table 2) with three regression algorithms: ridge regression, random forest regression, and support vector machine regression with RBF kernel, all implemented in Scikit-Learn [12] (additional results for boosted stumps and linear SVM in the appendix). We randomly split each dataset into three fixed parts: a training set, a validation set on which we optimized the hyper-parameters for all algorithms, and a testing set. We impute any missing values by the corresponding mean, and normalize the data to have zero mean and variance 1 on the training set. See the supplementary material for more experimental details.

To compare two models by computing the WLR on a reasonable number of diffs, we have made the testing sets as large as possible, so that the expected number of diffs between two different models is large enough to derive accurate and statistically significant conclusions. Lastly, we note that the churn metric does not require labels, so it can be computed on an unlabeled dataset.

### 5.1 Experimental Set-up and Metrics

We assume an initial classifier is to be trained on $T_A$, and a later candidate trained on $T_B$ will be tested against the initial classifier. For the baseline of our experiments, we train classifier $A$ on $T_A$ and classifier $B$ on $T_B$ independently and without any stabilization, as shown in Figure 1.

For the RCP operator comparison, we train $A$ on $T_A$, then train $B^+ = \mathcal{S}(A, T_B)$. For the MCMC operator comparison, we run the MCMC chain for $k = 30$ steps—empirically enough for convergence

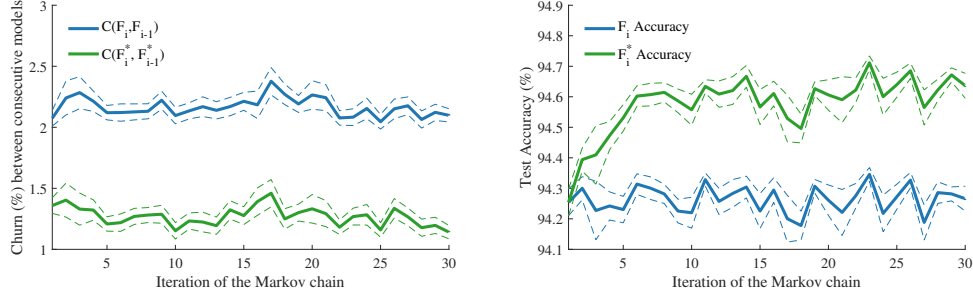

Figure 2: Left: Churn between consecutive models during the MCMC run on Nomao Dataset, with and without stabilization. Right: Accuracy of the intermediate models, with and without stabilization. Values are averaged over 40 runs of the chain. Dotted lines show standard errors.

for the datasets we considered as seen in Figure 2—and set $A^* = \mathcal{S}(F_k^*, T_A)$ and $B^* = \mathcal{S}(A^*, T_A)$. The dataset perturbation sub-samples $80\%$ of the examples in $T_A$ and randomly drops 3-7 features.

We run 40 independent chains to measure the variability, and report the average outcome and standard deviation. Figure 2 (left) plots the average and standard deviation of the churn along the 40 traces, and Figure 2 (right) shows the accuracy.

For each experiment we report the churn ratio $C_r$ between the initial classifier and candidate change, that is, $C_r = C(B^+, A)/C(B, A)$ for the RCP operator, and $C_r = C(B^*, A^*)/C(B, A)$ for the MCMC operator, and $C_r = C(B, A)/C(B, A) = 1$ for the baseline experiment. The most important metric in practice is how easy it is to tell if $B$ is an improvement over $A$, which we quantify by the WLR between the candidate and initial classifier for each experiment. To help interpret the WLR, we also report the resulting probability $p_{win}$ that we would conclude that the candidate change is positive ($p \leq 0.05$) with a random 100-example set of differences.

Lastly, to demonstrate that the proposed methods reduce the churn without adversely impacting the accuracy of the models, we also report the accuracy of the different trained models for a large test set, though the point of this work is that a sufficiently-large labeled test set may not be available in a real setting (see Section 1.1), and note that even if available, using a fixed test set to test many different changes will lead to overfitting.

## 5.2 Results

Table 3 shows results using reasonable default values of $\alpha = 0.5$ and $\epsilon = 0.5$ for both RCP and the MCMC (for results with other values of $\alpha$ and $\epsilon$ see Appendix D). As seen in the $C_r$ rows of the table, RCP reduces churn over the baseline in all 9 cases, generally by $20\%$, but as much as $46\%$ for ridge regression on the Nomao dataset. Similarly, running RCP in the Markov Chain also reduces the churn compared to the baseline in all 9 cases, and by slightly more on average than with the one-step RCP.

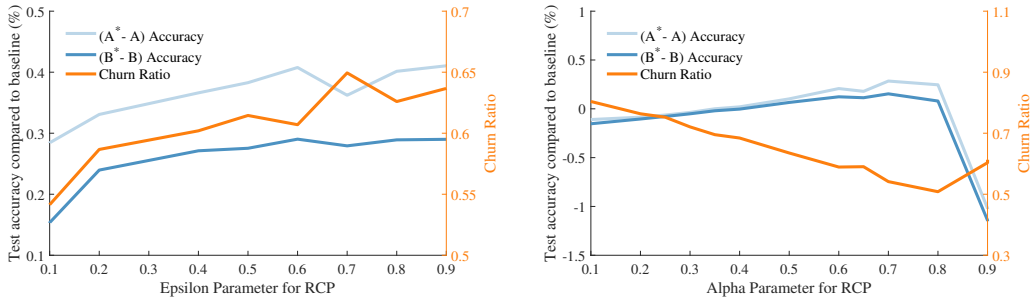

Figure 3: SVM on Nomao dataset. Left: Testing accuracy of $A^*$ and $B^*$ compared to $A$ and $B$, and churn ratio $C_r$ as a function of $\epsilon$, for fixed $\alpha = 0.7$. Both the accuracy and the churn ratio tend to increase with larger values of $\epsilon$. Right: Accuracies and the churn ratio versus $\alpha$, for fixed $\epsilon = 0.1$. There is a sharp decrease in accuracy with $\alpha > 0.8$ likely due to divergence in the chain.

Table 3: Experiment results on 3 domains with 3 different training algorithms for a single step RCP and the MCMC methods. For the MCMC experiment, we report the numbers with the standard deviation over the 40 runs of the chain.

| | | | Baseline No Stabilization | RCP $\alpha=0.5, \epsilon=0.5$ | MCMC, $k=30$ $\alpha=0.5, \epsilon=0.5$ |
|---|---|---|---|---|---|
| Nomao | Ridge | WLR | 1.24 | 1.40 | 1.31 |
| | | $p_{\text{win}}$ | 26.5 | 49.2 | 36.5 |
| | | $C_r$ | 1.00 | 0.54 | $0.54 \pm 0.06$ |
| | | Acc $V_1$ / $V_2$ | 93.1 / 93.4 | 93.1 / 93.4 | $93.2 \pm 0.1$ / $93.4 \pm 0.1$ |
| | RF | WLR | 1.02 | 1.13 | 1.09 |
| | | $p_{\text{win}}$ | 5.6 | 13.4 | 9.8 |
| | | $C_r$ | 1.00 | 0.83 | $0.83 \pm 0.05$ |
| | | Acc $V_1$ / $V_2$ | 94.8 / 94.8 | 94.8 / 95.0 | $94.9 \pm 0.2$ / $95.0 \pm 0.2$ |
| | SVM | WLR | 1.70 | 2.51 | 2.32 |
| | | $p_{\text{win}}$ | 82.5 | 99.7 | 99.2 |
| | | $C_r$ | 1.00 | 0.75 | $0.69 \pm 0.06$ |
| | | Acc $V_1$ / $V_2$ | 94.6 / 95.1 | 94.6 / 95.2 | $94.8 \pm 0.2$ / $95.3 \pm 0.1$ |
| News | Ridge | WLR | 0.95 | 0.94 | 1.04 |
| | | $p_{\text{win}}$ | 2.5 | 2.4 | 6.7 |
| | | $C_r$ | 1.00 | 0.75 | $0.78 \pm 0.04$ |
| | | Acc $V_1$ / $V_2$ | 65.1 / 65.0 | 65.1 / 65.0 | $65.0 \pm 0.1$ / $65.1 \pm 0.1$ |
| | RF | WLR | 1.07 | 1.02 | 1.10 |
| | | $p_{\text{win}}$ | 8.5 | 5.7 | 10.8 |
| | | $C_r$ | 1.00 | 0.69 | $0.67 \pm 0.04$ |
| | | Acc $V_1$ / $V_2$ | 64.5 / 65.1 | 64.5 / 64.7 | $64.3 \pm 0.3$ / $64.8 \pm 0.2$ |
| | SVM | WLR | 1.17 | 1.26 | 1.24 |
| | | $p_{\text{win}}$ | 18.4 | 29.4 | 26.1 |
| | | $C_r$ | 1.00 | 0.77 | $0.86 \pm 0.02$ |
| | | Acc $V_1$ / $V_2$ | 64.9 / 65.4 | 64.9 / 65.4 | $64.8 \pm 0.1$ / $65.4 \pm 0.1$ |
| Twitter Buzz | Ridge | WLR | 1.71 | 3.54 | 1.53 |
| | | $p_{\text{win}}$ | 83.1 | 100.0 | 66.4 |
| | | $C_r$ | 1.00 | 0.85 | $0.65 \pm 0.05$ |
| | | Acc $V_1$ / $V_2$ | 89.7 / 89.9 | 89.7 / 90.0 | $90.1 \pm 0.1$ / $90.2 \pm 0.1$ |
| | RF | WLR | 1.35 | 1.15 | 1.15 |
| | | $p_{\text{win}}$ | 41.5 | 16.1 | 15.9 |
| | | $C_r$ | 1.00 | 0.86 | $0.77 \pm 0.07$ |
| | | Acc $V_1$ / $V_2$ | 96.2 / 96.4 | 96.2 / 96.3 | $96.3 \pm 0.1$ / $96.3 \pm 0.1$ |
| | SVM | WLR | 1.35 | 1.77 | 1.55 |
| | | $p_{\text{win}}$ | 42.2 | 86.6 | 68.4 |
| | | $C_r$ | 1.00 | 0.70 | $0.70 \pm 0.03$ |
| | | Acc $V_1$ / $V_2$ | 96.0 / 96.1 | 96.0 / 96.1 | $96.1 \pm 0.1$ / $96.2 \pm 0.1$ |

In some cases, the reduced churn has a huge impact on the WLR. For example, for the SVM on Twitter, the 30% churn reduction by RCP raised the WLR from 1.35 to 1.77, making it twice as likely that labelling 100 differences would have verified the change was good (compare $p_{win}$ values). MCMC provides a similar churn reduction, but the WLR increase is not as large.

In addition to the MCMC providing slightly more churn reduction on average than RCP, running the Markov chain provides slightly higher accuracy on average as well, most notably for the ridge classifier on the Twitter dataset, raising initial classifier accuracy by 2.3% over the baseline. We hypothesize this is due to the regularization effect of the perturbed training during the MCMC run, resembling the effect of dropout in neural networks.

We used fixed values of $\alpha = 0.5$ and $\epsilon = 0.5$ for all the experiments in Table 3, but note that results will vary with the choice of $\alpha$ and $\epsilon$, and if they can be tuned with cross-validation or otherwise, results can be substantially improved. Figure 3 illustrates the dependence on these hyper-parameters: the left plot shows that small values of $\epsilon$ result in lower churn with reduced improvement on accuracy, and the right plot shows that increasing $\alpha$ reduces churn, and also helps increase accuracy, but at values larger than 0.8 causes the Markov chain to diverge.

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
