[Supplementary Material]

# Launch and Iterate: Reducing Prediction Churn Appendix

**Q. Cormier**
ENS Lyon
15 parvis René Descartes
Lyon, France
quentin.cormier@ens-lyon.fr

**M. Milani Fard, K. Canini, M. R. Gupta**
Google Inc.
1600 Amphitheatre Parkway
Mountain View, CA 94043
{mmilanifard,canini,mayagupta}@google.com

This appendix includes proofs of the theorems presented in the paper and details about the experimental design.

## A  Proof of Theorem 1

*Proof.* Let $z = (x, y)$ be a fixed testing sample. We define the function $G_z : (\mathcal{X} \times \mathcal{Y})^{2m} \to \mathbb{R}$ as:

$$G_z(T, T') = \ell_\gamma(f_T(x), y) - \ell_\gamma(f_{T'}(x), y). \tag{1}$$

For any $i \in \{1, .., m\}$, if we re-sample the $i$th element in $T$ to get $T^i$, using the $\beta$-stability of the learning algorithm and Lipschitz continuity of $\ell_\gamma$ we get:

$$|G_z(T, T') - G_z(T^i, T')| \le \frac{1}{\gamma}|f_T(x) - f_{T^i}(x)| \le \frac{\beta}{\gamma}. \tag{2}$$

The same inequality holds for $|G_z(T, T') - G_z(T, T'^i)|$. We have $\mathbb{E}_{T,T' \sim D^m}[G_z(T, T')] = 0$, and thus can apply the McDiarmid inequality to get:

$$\Pr_{T,T' \sim D^m}[|G_z(T, T')| > \epsilon] \le 2e^{-\frac{\epsilon^2 \gamma^2}{m\beta^2}}. \tag{3}$$

Integrating the above gives us the bound over the expectation:

$$\mathbb{E}_{T,T' \sim \mathcal{D}^m}[|G_z(T, T')|] \le \int_0^\infty \Pr_{T,T' \sim \mathcal{D}^m}[|G_z(T, T')| > \epsilon]d\epsilon \le \frac{\beta\sqrt{\pi m}}{\gamma}. \tag{4}$$

The above inequality holds for any fixed $z$ and thus holds for the expectation:

$$
\begin{aligned}
\mathbb{E}_{T,T' \sim \mathcal{D}^m}[C_\gamma(f_1, f_2)] &= \mathbb{E}_{T,T' \sim \mathcal{D}^m}\left[\mathbb{E}_{Z \sim \mathcal{D}}[|G_Z(T, T')|]\right] &(5)\\
&= \mathbb{E}_{Z \sim \mathcal{D}}\left[\mathbb{E}_{T,T' \sim \mathcal{D}^m}[|G_Z(T, T')|]\right] &(6)\\
&\le \frac{\beta\sqrt{\pi m}}{\gamma}. &(7)
\end{aligned}
$$

$\square$

## B  Proof of Theorem 2

*Proof.* We again use the McDiarmid inequality, on the function $H : (\mathcal{X} \times \mathcal{Y})^{2m} \to \mathbb{R}$ defined as:

$$H(T, T') = C_\gamma(f_T, f_{T'}) = \mathbb{E}_{(X,Y) \sim \mathcal{D}}[|\ell_\gamma(f_T(X), Y) - \ell_\gamma(f_{T'}(X), Y)|]. \tag{8}$$

For any $i \in \{1, .., m\}$, if we re-sample the $i$th element in $T$ to get $T^i$, using the $\beta$-stability of the learning algorithm and Lipschitz continuity of $\ell_\gamma$ we get:

$$
\begin{aligned}
|H(T,T') - H(T^i, T')| \quad &\leq \quad \mathop{\mathbb{E}}_{(X,Y)\sim\mathcal{D}} \Big[ ||\ell_\gamma(f_T(X),Y) - \ell_\gamma(f_{T'}(X),Y)| - && (9) \\
&\qquad\qquad |\ell_\gamma(f_{T^i}(X),Y) - \ell_\gamma(f_{T'}(X),Y)|| \Big] && (10) \\
&\leq \quad \mathop{\mathbb{E}}_{(X,Y)\sim\mathcal{D}} [|\ell_\gamma(f_T(X),Y) - \ell_\gamma(f_{T^i}(X),Y)|] && (11) \\
&\leq \quad \frac{\beta}{\gamma}, && (12)
\end{aligned}
$$

where line (11) is by reverse triangular inequality. Same bound similarly holds for replacing the $i$th element in $T'$: $|H(T,T') - H(T,T'^i)| \leq \beta/\gamma$. Applying McDiarmid inequality and using the bound on the expectation of $H$ from Theorem 1 completes the proof:

$$
\begin{aligned}
\mathop{\Pr}_{T,T'\sim\mathcal{D}^m} \left\{ C_\gamma(f_T, f_{T'}) > \epsilon + \frac{\sqrt{\pi m}\beta}{\gamma} \right\} \quad &\leq \quad \mathop{\Pr}_{T,T'\sim\mathcal{D}^m} \left\{ H(T,T') > \epsilon + \mathop{\mathbb{E}}_{T,T'\sim\mathcal{D}^m}[H(T,T')] \right\} \\
&\leq \quad e^{-\frac{\epsilon^2\gamma^2}{m\beta^2}}. && (13)
\end{aligned}
$$

$\square$

## C   Proof of Theorem 3

The proof partly follows Lemma 21 from [1] and Theorem 4 from [2]. Define:

$$
\begin{aligned}
\ell_j(g) &= (g(x_j) - y_j)^2 && (14) \\
\hat{R}_T(g) &= \frac{1}{m} \sum_{j=1}^{m} \ell_j(g) && (15) \\
\hat{R}_T^{\backslash i}(g) &= \frac{1}{m} \sum_{\substack{j=1 \\ j \neq i}}^{m} \ell_j(g) && (16) \\
R_T(g) &= \hat{R}_T(g) + \lambda\|g\|_k^2 && (17) \\
R_T^{\backslash i}(g) &= \hat{R}_T^{\backslash i}(g) + \lambda\|g\|_k^2. && (18)
\end{aligned}
$$

By the assumption of the theorem, $f_T$ is the minimizer of $R_T$. Let $f_T^{\backslash i}$ be the minimizer of $R_T^{\backslash i}$.

**Lemma 1.** *With the assumptions of Theorem 3, we have for all $i$:*

$$
\forall x : (f_T(x) - f_T^{\backslash i}(x))^2 \leq \frac{\kappa^4}{\lambda^2 m^2}(f_T(x_i) - y_i)^2. \tag{19}
$$

*Proof of Lemma 1.* To simply the notation, we drop the $T$ subscript throughout the proof of this lemma. Let $d_\phi(f,g)$ be the functional Bregman divergence [3]:

$$
d_\phi(f,g) = \phi(f) - \phi(g) - \nabla\phi(g; f-g), \tag{20}
$$

where $\nabla\phi(g; .)$ is the Fréchet derivative of $\phi$ at $g$. Since $f$ and $f^{\backslash i}$ are minimizers of $R$ and $R^{\backslash i}$ respectively, we have: $\nabla R(f; .) = 0$ and $\nabla R^{\backslash i}(f^{\backslash i}; .) = 0$. We thus have:

$$
\begin{aligned}
d_R(f^{\backslash i}, f) + d_{R^{\backslash i}}(f, f^{\backslash i}) &= R(f^{\backslash i}) - R(f) + R^{\backslash i}(f) - R^{\backslash i}(f^{\backslash i}) && (21) \\
&= \frac{1}{m}\ell_i(f^{\backslash i}) - \frac{1}{m}\ell_i(f), && (22)
\end{aligned}
$$

where the last line follows by the definition of $R$ and $R^{\setminus i}$. By non-negativity and additivity of divergence ($d_{A+B} = d_A + d_B$) we have:

$$0 \quad \leq \quad d_{\hat{R}^{\setminus i}}(f, f^{\setminus i}) + d_{\hat{R}^{\setminus i}}(f^{\setminus i}, f) \tag{23}$$

$$= \quad -\lambda d_{\|\cdot\|_k^2}(f, f^{\setminus i}) - \lambda d_{\|\cdot\|_k^2}(f^{\setminus i}, f) + d_{R^{\setminus i}}(f, f^{\setminus i}) + d_{R^{\setminus i}}(f^{\setminus i}, f) \tag{24}$$

$$= \quad -\lambda d_{\|\cdot\|_k^2}(f, f^{\setminus i}) - \lambda d_{\|\cdot\|_k^2}(f^{\setminus i}, f) + d_{R^{\setminus i}}(f, f^{\setminus i}) + d_R(f^{\setminus i}, f) - \frac{1}{m} d_{\ell_i}(f^{\setminus i}, f) \tag{25}$$

$$= \quad -\lambda d_{\|\cdot\|_k^2}(f, f^{\setminus i}) - \lambda d_{\|\cdot\|_k^2}(f^{\setminus i}, f) + \frac{1}{m}\ell_i(f^{\setminus i}) - \frac{1}{m}\ell_i(f) - \frac{1}{m} d_{\ell_i}(f^{\setminus i}, f) \tag{26}$$

$$= \quad -\lambda d_{\|\cdot\|_k^2}(f, f^{\setminus i}) - \lambda d_{\|\cdot\|_k^2}(f^{\setminus i}, f) + \frac{1}{m}\nabla \ell_i(f; f^{\setminus i} - f), \tag{27}$$

where line (26) is by the derivation in line (22), and line (27) is by the definition of the Bregman divergence. In the RKHS space, we have $d_{\|\cdot\|_k^2}(g, g') = \|g - g'\|_k^2$, and by assumption of Theorem 3 we have $\forall x : |g(x)| \leq \kappa \|g\|_k$. Substituting the Fréchet derivative in the above inequality, we get:

$$\|f - f^{\setminus i}\|_k^2 \quad \leq \quad \frac{1}{\lambda m}(f^{\setminus i}(x) - f(x))(f(x_i) - y_i) \tag{28}$$

$$\leq \quad \frac{\kappa}{\lambda m}\|f^{\setminus i} - f\|_k (f(x_i) - y_i). \tag{29}$$

Cancelling the sides and squaring both sides, we get for all $x$:

$$(f(x) - f^{\setminus i}(x))^2 \quad \leq \quad \kappa^2 \|f - f^{\setminus i}\|_k^2 \tag{30}$$

$$\leq \quad \frac{\kappa^4}{\lambda^2 m^2}(f(x_i) - y_i)^2. \tag{31}$$

$\square$

*Proof of Theorem 3.* Let $V = \ell_\gamma(f_T(X), Y) - \ell_\gamma(f_{T'}(X), Y)$. Define $V_i$, $1 \leq i \leq 2m$ as:

$$V_i = \begin{cases} \ell_\gamma(f_T^{\setminus i}(X), Y) - \ell_\gamma(f_{T'}(X), Y) & \text{if } i \leq m \\ \ell_\gamma(f_T(X), Y) - \ell_\gamma(f_{T'}^{\setminus(i-m)}(X), Y) & \text{if } i > m \end{cases} \tag{32}$$

It is easy to see that $\mathbb{E}_{T,T' \sim \mathcal{D}^m}[V] = 0$. Using the concentration inequality of Theorem 6 from [4] on $V$ and $V_i$, the symmetry of the training algorithm, and the symmetry of $V$ on $T$ and $T'$ we get:

$$\underset{\substack{T,T' \sim \mathcal{D}^m \\ (X,) \sim \mathcal{D}}}{\mathbb{E}}[(\ell_\gamma(f_T(X), Y) - \ell_\gamma(f_{T'}(X), Y))^2] \quad = \quad \underset{\substack{T,T' \sim \mathcal{D}^m \\ (X,Y) \sim \mathcal{D}}}{\mathbb{V}\text{ar}}[V] \tag{33}$$

$$\leq \quad \sum_{i=1}^{2m} \underset{\substack{T,T' \sim \mathcal{D}^m \\ (X,Y) \sim \mathcal{D}}}{\mathbb{E}}[(V - V_i)^2] \tag{34}$$

$$= \quad 2 \underset{\substack{T,T' \sim \mathcal{D}^m \\ (X,Y) \sim \mathcal{D}}}{\mathbb{E}}\left[\sum_{i=1}^{m}(V - V_i)^2\right] \tag{35}$$

$$= \quad \frac{2}{\gamma^2} \underset{\substack{T,T' \sim \mathcal{D}^m \\ (X,Y) \sim \mathcal{D}}}{\mathbb{E}}\left[\sum_{i=1}^{m}(f_T(X) - f_T^{\setminus i}(X))^2\right], \tag{36}$$

where line (36) is by Lipschitz continuity of $\ell_\gamma$. Applying Lemma 1 to RHS completes the proof. $\square$

# D    Further Experimental Details

Table 1 includes further details on the datasets used for experiments presented in the paper.

Table 1: Full details of the datasets used in the experimental analysis.

|  | Nomao [5] | News Popularity [6] | Twitter Buzz [7] |
|---|---|---|---|
| # Features | 89 continuous<br>31 nominal<br>some missing values | 61 features<br>no missing values | 77 features<br>evolution of 11 primatry<br>features through time<br>no missing values |
| # Samples | 34,465 | 39,797 | Sub-sampled 46,902 |
| Goal | predict if two business entities are the same | predict if a news will be shared more than 1400 times | predict if a tweet is going to be popular |
| $T_A$ | 4000 samples<br>drop first 5 features | 8000 samples<br>drop the 3 features:<br>*self_reference_min*<br>*self_reference_max*<br>*self_reference_avg* | 4000 samples<br>drop last 7 features |
| $T_B$ | 5000 samples<br>all the features | 10000 samples<br>all the features | 5000 samples<br>all the features |
| Validation Set | 1000 samples | 1000 samples | 1000 samples |
| Testing Set | 28465 samples | 28797 samples | 45402 samples |

We optimized the hyper-parameters of each algorithm for each datasets on the validation set. Details of the chosen hyper-parameters for each algorithm is included in Table 2. The names of the parameters match the names used in Scikit-Learn [8].

Table 2: We summarize here the regularization parameters used to train the models. These parameters have been selected using a validation set of 1000 samples.

|  | Ridge $\alpha$ | RFT-Regression min_weight_fraction_leaf | SVM $C$ | Adaboost learning_rate | LinearSVR $C$ |
|---|---|---|---|---|---|
| Nomao | 0.02 | 0.0001 | 10 | 1.5 | 0.5 |
| News | 2 | 0.01 | 1.5 | 5 | 10 |
| Twitter-Buzz | 1 | 0.002 | 50 | 1.0 | 75 |

Full results for all experiments are included in Table 3. We have included further results on linear SVM and AdaBoost (boosted stumps). However, note that there is a regression in accuracy between the two versions of the model for the baseline algorithm. We believe that that our hyper-parameter optimization did not find a good solution for these algorithms (likely resulting in over-fitting), or that we could not effectively use the implementation in Scikit-Learn [8].

Table 3: Experiment results on 3 domains with 5 different training algorithms for a single step RCP and the MCMC methods. For the MCMC experiment, we report the numbers with the standard deviation over the 40 runs of the chain.

| | | | Baseline No RCP, No Chain | RCP $\alpha = 0.5, \epsilon = 0.5$ | MCMC, $k = 30$ $\alpha = 0.5, \epsilon = 0.5$ | MCMC, $k = 30$ $\alpha = 0.7, \epsilon = 0.1$ |
|---|---|---|---|---|---|---|
| Nomao | Ridge | WLR | 1.24 | 1.40 | 1.31 | 1.60 |
| | | $p_{\text{win}}$ | 26.5 | 49.2 | 36.5 | 73.9 |
| | | $C_r$ | 1.00 | 0.54 | $0.54 \pm 0.06$ | $0.32 \pm 0.05$ |
| | | Acc $V_1$ / $V_2$ | 93.1 / 93.4 | 93.1 / 93.4 | $93.2 \pm 0.1$ / $93.4 \pm 0.1$ | $93.0 \pm 0.3$ / $93.2 \pm 0.2$ |
| | RF | WLR | 1.02 | 1.13 | 1.09 | 1.12 |
| | | $p_{\text{win}}$ | 5.6 | 13.4 | 9.8 | 13.1 |
| | | $C_r$ | 1.00 | 0.83 | $0.83 \pm 0.05$ | $0.59 \pm 0.05$ |
| | | Acc $V_1$ / $V_2$ | 94.8 / 94.8 | 94.8 / 95.0 | $94.9 \pm 0.2$ / $95.0 \pm 0.2$ | $94.7 \pm 0.2$ / $94.8 \pm 0.2$ |
| | AdaBoost | WLR | 0.79 | 0.79 | 0.00 | 0.00 |
| | | $p_{\text{win}}$ | 0.2 | 0.2 | 0.0 | 0.0 |
| | | $C_r$ | 1.00 | 1.00 | $0.01 \pm 0.06$ | $0.00 \pm 0.00$ |
| | | Acc $V_1$ / $V_2$ | 85.7 / 85.3 | 85.7 / 85.3 | $75.7 \pm 2.4$ / $75.6 \pm 2.2$ | $77.4 \pm 6.2$ / $77.4 \pm 6.2$ |
| | LinSVM | WLR | 0.64 | 0.89 | 0.90 | 2.60 |
| | | $p_{\text{win}}$ | 0.0 | 1.2 | 1.3 | 99.8 |
| | | $C_r$ | 1.00 | 0.75 | $0.76 \pm 0.02$ | $0.22 \pm 0.02$ |
| | | Acc $V_1$ / $V_2$ | 90.1 / 86.2 | 90.1 / 89.3 | $90.1 \pm 0.4$ / $89.4 \pm 0.3$ | $90.1 \pm 0.5$ / $91.9 \pm 0.5$ |
| | SVM | WLR | 1.70 | 2.51 | 2.32 | 2.08 |
| | | $p_{\text{win}}$ | 82.5 | 99.7 | 99.2 | 97.1 |
| | | $C_r$ | 1.00 | 0.75 | $0.69 \pm 0.06$ | $0.54 \pm 0.03$ |
| | | Acc $V_1$ / $V_2$ | 94.6 / 95.1 | 94.6 / 95.2 | $94.8 \pm 0.2$ / $95.3 \pm 0.1$ | $94.9 \pm 0.2$ / $95.2 \pm 0.1$ |
| News | Ridge | WLR | 0.95 | 0.94 | 1.04 | 0.97 |
| | | $p_{\text{win}}$ | 2.5 | 2.4 | 6.7 | 3.4 |
| | | $C_r$ | 1.00 | 0.75 | $0.78 \pm 0.04$ | $0.42 \pm 0.06$ |
| | | Acc $V_1$ / $V_2$ | 65.1 / 65.0 | 65.1 / 65.0 | $65.0 \pm 0.1$ / $65.1 \pm 0.1$ | $64.7 \pm 0.2$ / $64.7 \pm 0.2$ |
| | RF | WLR | 1.07 | 1.02 | 1.10 | 1.24 |
| | | $p_{\text{win}}$ | 8.5 | 5.7 | 10.8 | 26.6 |
| | | $C_r$ | 1.00 | 0.69 | $0.67 \pm 0.04$ | $0.04 \pm 0.04$ |
| | | Acc $V_1$ / $V_2$ | 64.5 / 65.1 | 64.5 / 64.7 | $64.3 \pm 0.3$ / $64.8 \pm 0.2$ | $63.0 \pm 0.4$ / $63.0 \pm 0.4$ |
| | AdaBoost | WLR | 0.72 | 0.72 | 0.81 | 0.00 |
| | | $p_{\text{win}}$ | 0.0 | 0.0 | 0.3 | 0.0 |
| | | $C_r$ | 1.00 | 1.00 | $7.88 \pm 12.07$ | $0.03 \pm 0.06$ |
| | | Acc $V_1$ / $V_2$ | 59.3 / 59.2 | 59.3 / 59.2 | $59.4 \pm 0.2$ / $59.2 \pm 0.0$ | $58.7 \pm 1.1$ / $58.7 \pm 1.1$ |
| | LinSVM | WLR | 0.81 | 1.24 | 1.03 | 1.02 |
| | | $p_{\text{win}}$ | 0.3 | 26.4 | 6.1 | 5.3 |
| | | $C_r$ | 1.00 | 0.90 | $1.10 \pm 0.19$ | $1.12 \pm 0.26$ |
| | | Acc $V_1$ / $V_2$ | 63.3 / 62.5 | 63.3 / 64.1 | $63.5 \pm 0.5$ / $63.6 \pm 0.5$ | $63.0 \pm 0.8$ / $63.1 \pm 0.7$ |
| | SVM | WLR | 1.17 | 1.26 | 1.24 | 1.25 |
| | | $p_{\text{win}}$ | 18.4 | 29.4 | 26.1 | 28.0 |
| | | $C_r$ | 1.00 | 0.77 | $0.86 \pm 0.02$ | $0.61 \pm 0.02$ |
| | | Acc $V_1$ / $V_2$ | 64.9 / 65.4 | 64.9 / 65.4 | $64.8 \pm 0.1$ / $65.4 \pm 0.1$ | $64.7 \pm 0.2$ / $65.1 \pm 0.1$ |
| Twitter Buzz | Ridge | WLR | 1.71 | 3.54 | 1.53 | 1.58 |
| | | $p_{\text{win}}$ | 83.1 | 100.0 | 66.4 | 71.9 |
| | | $C_r$ | 1.00 | 0.85 | $0.65 \pm 0.05$ | $0.44 \pm 0.04$ |
| | | Acc $V_1$ / $V_2$ | 89.7 / 89.9 | 89.7 / 90.0 | $90.1 \pm 0.1$ / $90.2 \pm 0.1$ | $89.7 \pm 0.1$ / $89.7 \pm 0.1$ |
| | RF | WLR | 1.35 | 1.15 | 1.15 | 1.03 |
| | | $p_{\text{win}}$ | 41.5 | 16.1 | 15.9 | 6.0 |
| | | $C_r$ | 1.00 | 0.86 | $0.77 \pm 0.07$ | $0.42 \pm 0.10$ |
| | | Acc $V_1$ / $V_2$ | 96.2 / 96.4 | 96.2 / 96.3 | $96.3 \pm 0.1$ / $96.3 \pm 0.1$ | $96.2 \pm 0.1$ / $96.2 \pm 0.1$ |
| | AdaBoost | WLR | 0.93 | 0.90 | 1.13 | 1.17 |
| | | $p_{\text{win}}$ | 1.8 | 1.2 | 13.3 | 18.4 |
| | | $C_r$ | 1.00 | 1.03 | $0.80 \pm 0.18$ | $0.22 \pm 0.07$ |
| | | Acc $V_1$ / $V_2$ | 95.0 / 95.0 | 95.0 / 95.0 | $94.2 \pm 0.4$ / $94.2 \pm 0.4$ | $95.5 \pm 0.3$ / $95.5 \pm 0.3$ |
| | LinSVM | WLR | 0.22 | 2.66 | 3.71 | 3.82 |
| | | $p_{\text{win}}$ | 0.0 | 99.9 | 100.0 | 100.0 |
| | | $C_r$ | 1.00 | 0.52 | $0.61 \pm 0.45$ | $0.41 \pm 0.22$ |
| | | Acc $V_1$ / $V_2$ | 94.8 / 91.2 | 94.8 / 96.2 | $92.2 \pm 2.7$ / $92.7 \pm 2.5$ | $93.0 \pm 2.0$ / $93.2 \pm 2.0$ |
| | SVM | WLR | 1.35 | 1.77 | 1.55 | 1.33 |
| | | $p_{\text{win}}$ | 42.2 | 86.6 | 68.4 | 39.3 |
| | | $C_r$ | 1.00 | 0.70 | $0.70 \pm 0.03$ | $0.50 \pm 0.03$ |
| | | Acc $V_1$ / $V_2$ | 96.0 / 96.1 | 96.0 / 96.1 | $96.1 \pm 0.1$ / $96.2 \pm 0.1$ | $96.1 \pm 0.1$ / $96.2 \pm 0.1$ |

# E  Link between the accuracies, the WLR, and the Churn

Given two classifiers $f_A$ and $f_B$ (that is, any measurable function from $\mathbb{R}^d$ to $\{-1, 1\}$), we define the Win/Loss Ratio (*WLR*) to be $\frac{p}{1-p}$ with $p = \Pr[f_B(X) = Y \mid f_B(X) \neq f_A(X)]$: $p$ is the probability that model $f_B$ is correct knowing that models $f_A$ and $f_B$ are giving a different answer.

Recall that the Churn between $f_A$ and $f_B$ is defined to be:

$$C = \Pr[f_A(X) \neq f_B(X)],$$

and that the accuracies of $f_A$ and $f_B$ are given by:

$$Acc_A = \Pr[f_A(X) = Y], \ Acc_B = \Pr[f_B(X) = Y].$$

**Lemma 2.** *The relation between the $WLR$, Churn and accuracies of the classifiers $f_A$ and $f_B$ is the following:*

$$WLR = \frac{p}{1-p}, \ p = \frac{1}{2} + \frac{Acc_B - Acc_A}{2C}. \tag{37}$$

*Proof.*

$$Acc_B = \Pr[f_B(X) = Y, f_B(X) \neq f_A(X)] + \Pr[f_B(X) = Y, \ f_B(X) = f_A(X)].$$

Futhermore:

$$\Pr[f_B(X) = Y, \ f_B(X) = f_A(X)] = \Pr[f_A(X) = Y, \ f_B(X) = f_A(X)]$$
$$= Acc_A - \Pr[f_A(X) = Y, f_A(X) \neq f_B(X)].$$

Thus using the Bayes' theorem we deduce that:

$$Acc_B = pC + Acc_A - (1-p)C,$$

which gives the result. $\qquad\square$

As discussed before, (37) confirms that for a fixed accuracy gain, less Churn is better as it increases $p$, and thus increases the WLR.

A statistical hypothesis test is usually used to decide if model $f_B$ is statistically significantly better than $f_A$. In this setting, increasing the ratio $\frac{Acc_B - Acc_A}{C}$ increases uniformly the power of such test.