[Reviews · NeurIPS 2016]

Reviewer 1

Summary

The paper is concerned with what the authors call churn, and proposes to regularize classifiers in a specific way to reduce it.

Qualitative Assessment

This could be a novel, interesting idea with the potential to generate new research directions. It is difficult for me to judge. The idea that "churn" is undesirable is reasonable, and the regularization idea seems an appropriate way to prevent excessive churn. The theoretical results do not appear to tell us much about how to choose alpha and epsilon in practice. I do not understand why the authors call the method in Section 2.2 a Markov chain Monte Carlo instead of just a Markov chain. The results are good.

Confidence in this Review

2-Confident (read it all; understood it all reasonably well)


Reviewer 2

Summary

This paper takes a stab at providing a foundation to analyze the churn of ML models: e.g., how much one model trained as an improvement to another changes the predictions of the model. The more changes, the more difficult to evaluate in practice. The authors show a simple method to reduce churn by regularizing towards past model's predictions, and provide a theoretical analysis of this approach. I really like this paper because it is easy to read, very thought provoking, and potentially very useful in practice.

Qualitative Assessment

Major strengths of this paper: + Thorough, theoretical treatment of a problem that is super relevant to any practitioner of machine learning. + The experiments are thorough, and the P_win analysis is very nice. I really appreciate the thought that the authors put into making this more theoretical work appealing in a very practical way. + The connection to dropout, and the use of the Markov chain as away to get a robust model, is very interesting. + The gains in some cases are very big in the datasets used -- significantly improving churn while not affecting or improving accuracy. Major weaknesses: - Unfortunately, one major take-away here isn't particularly inspiring: e.g. there's a trade-off between churn and accuracy. - Also, the thought of having to train 30-40 models to burn in in order to test this approach isn't particularly appealing. Another interesting direction for dealing with churn could be unlabelled data, or applying via constraints: e.g. if we are willing to accep X% churn, and have access to unlabeled target data, what's the best way to use that to improve the stability of our model?

Confidence in this Review

2-Confident (read it all; understood it all reasonably well)


Reviewer 3

Summary

This paper presents methods aimed at reducing the prediction churn (changes in the predictor) without sacrificing the potential gain in accuracy that a change in the predictor could introduce. After well motivating the problem, the paper proposes two stabilization operators that can be used in conjonction with a MCMC chain that iteratively perturb the training sets; hopefully mimicking the changes that could occur in practice as more data is gathered and new features are added. Theoretical results (in a restricted setting) are presented that support the proposed method along with experiments on real data sets.

Qualitative Assessment

The paper is well presented and the problem well motivated. The proposed stabilization operators make sens but introduce two hyperparameters to tune. To be effective, the proposed MCMC chain should mimick the changes that are likely to occur in practice: ie, the number of available training examples is (almost always) increasing with time and the number of features is usually increasing with time. Unfortunately, the proposed MCMC chain keeps both of these numbers constant (or approximately constant) over time and this affects the potential impact score pretty much...

Confidence in this Review

2-Confident (read it all; understood it all reasonably well)


Reviewer 4

Summary

The paper articulates an important problem---reducing unnecessary changes in the predictions of machine learning algorithms, describes why this problem is relevant (it hinders usability of prediction, and hinders ability to statistically validate that the trained model improved), and gives this problem a name (“prediction churn”) and very general formal definition. The paper introduces a conceptually interesting and general fixed-point approach to reducing prediction churn, which it describes (in a somewhat hand-wavy way) as an MCMC technique. The approach combines regularization of new classifiers with respect to the previous classifier and training the new classifier on perturbed version of the training data set. Together, these techniques, when iterated, are expected to result in a final classifier that will not change much when re-trained (regularized) on the perturbed dataset when the perturbations are the result of real-world drift. The paper provides examples of two fairly general stability operators with hyperparameters governing the tradeoff between churn and accuracy. The paper provides some theoretical bounds on the “churn” for restricted settings, but does not provide theoretical characterization of the de-churning of the iterative Markov chain, such as when a stationary distribution exists. The paper includes experiments on real-world datasets and several learning algorithms that demonstrate that a one-step and Markov chain de-churning process successfully reduce churn while maintaining or improving accuracy. The experimental results do not emphasize the importance of the fixed-point property of the Markov chain.

Qualitative Assessment

The problem articulated and addressed by the paper is important, the techniques introduces are very general in their applicability, and the experimental results successfully demonstrate the utility of the techniques to addressing the problem. The fixed-point MCMC approach is very conceptually compelling, although it lacked theoretical analysis and rigor, and appeared to be not that significant to achieving good results (the results suggest a single iteration may be sufficient). After reading the highly suggestive section on the Markov chain, the theoretical results appeared underwhelming. A rigorous analysis of the Markov chain would have made the paper more self-contained and increased its technical strength. Alternatively, focusing on the single-step stabilization operator case, and obtaining more general theoretical results within that restricted setting, would have made the paper more self-contained and cohesive. The paper lacked polish, which made it harder to digest than necessary. For example, in the experiment section 5.1 the authors paper refers to "the RCP operator comparison" and "the MCMC operator comparison", and do not make it clear that the MCMC comparison itself is composed of multiple RCP operators. Issues of wording like this, as well as somewhat strange choices of order in which new notation is presented (e.g. T_A is presented on line 87 even though it is not necessary until later; the use of undefined notation P_T on line 91), reduced the clarity of the paper. It would have been interesting to see a relation of the technique to online Bayesian learning. The Markov chain, stabilization operator, and training data perturbation technique seem related to Bayesian approaches, and an analysis of the Markov chain (that was not included in the paper) may specifically benefit from this perspective. Finally, the paper would have benefited from a discussion or conclusion that discussed in more detail the potential impact of this work.

Confidence in this Review

1-Less confident (might not have understood significant parts)


Reviewer 5

Summary

This paper describes a method to reduce the net-zero win and loss changes (churn) of two models trained successively with increasing features and training size. A stabilization method is proposed to and used in a Markov chain to reduce churn. Both theoretical and empirical analysis is provided.

Qualitative Assessment

The paper uses a stabilization operator to regularize model B to be consistent with the previous model A. Theoretical and empirical results validate its reasonability. However, I'm a little bit doubting the necessity of such successive training method. The stabilization operator (e.g., RCP) essentially transfers information of model A which is trained on a different dataset T_A (e.g., less features) to model B. That is, now model B also "sees" the dataset T_A in addition to T_B. I am wondering what would happen if you directly train B on T_A+T_B, or more practically, on T_B with random perturbation (e.g., dropping features randomly) like dropout. Will this yield better accuracy and stability? Considering the recent success and popularity of such dropout-style methods, the authors should compare their method to it. The choice of perturbation P_T (Line.94) and the claim that classifier A* is "pre-trained to be robust to the kind of changes" (Line.104) are not that intuitive. Why A* is necessarily robust to the changes encoded in P_T (though I know with P_T F_k^* tends to generalize better)? How would it impact B*'s robustness? Please explain how the "True WLR Needed" in Table.1 is calculated. Line.178 re-labels -> re-label Table.3 is hard to read. It'd be nice to put different metrics (e.g., WLR) into separate tables to make comparison easier.

Confidence in this Review

2-Confident (read it all; understood it all reasonably well)


Reviewer 6

Summary

Machine learning algorithms involve many training iterations and constantly refining feature set, with an input stream of training examples. In such cases, we should be able to observe statistically significant improvements over the model in the previous iteration. The authors define a new term called "churn" which helps to determine whether a refined model (through more training) shows a statistically significant improvement over its previous version. The authors also provide a new training scheme which could be applied to most classifiers so as to reduce the churn (and thus we can be sure that the model has really improved). Through experiments, the authors show that they significantly reduce the value of the "churn" metric by using three different classifiers on three different datasets.

Qualitative Assessment

Pros: The formulation of the "churn" metric is useful especially when the classifier is trained on a constant stream of input data. Reducing the churn will ensure that additional resources spent contributed to significant improvement. In Table 3, what is V_1 and V_2. I assumed that they are the accuracies of the classifiers A and B. It would be helpful to discuss some intuitions behind why Eqn 2 and Eqn for Diplopia operator would reduce churn.

Confidence in this Review

1-Less confident (might not have understood significant parts)